# Graph Pooling via Ricci Flow

**Amy Feng**                                                                                    *afeng@college.harvard.edu*
*Harvard College*

**Melanie Weber**                                                                              *mweber@seas.harvard.edu*
*Harvard University*

**Reviewed on OpenReview:** *https://openreview.net/forum?id=xpBHp9WFvk*

## Abstract

Graph Machine Learning often involves the clustering of nodes based on similarity structure encoded in the graph's topology and the nodes' attributes. On homophilous graphs, the integration of pooling layers has been shown to enhance the performance of Graph Neural Networks by accounting for inherent multi-scale structure. Here, similar nodes are grouped together to coarsen the graph and reduce the input size in subsequent layers in deeper architectures. The underlying clustering approach can be implemented via graph pooling operators, which often rely on classical tools from Graph Theory. In this work, we introduce a graph pooling operator (ORC-Pool), which utilizes a characterization of the graph's geometry via Ollivier's discrete Ricci curvature and an associated geometric flow. Previous Ricci flow based clustering approaches have shown great promise across several domains, but are by construction unable to account for similarity structure encoded in the node attributes. However, in many ML applications, such information is vital for downstream tasks. ORC-Pool extends such clustering approaches to attributed graphs, allowing for the integration of geometric coarsening into Graph Neural Networks as a pooling layer.

## 1 Introduction

Multi-scale structure is ubiquitous in relational data across domains; examples include complex molecules in computational biology, systems of interacting particles in physics, as well as complex financial and social systems. Many graph learning tasks, such as clustering and coarsening, rely on such structure. Graph Neural Networks (GNNs) account for multi-scale structure via pooling layers, which form crucial building blocks in many state of the art architectures.

Node clustering often utilizes an implicit or explicit pooling operation, which decomposes a given graph into densely connected subgraphs by grouping similar nodes. Applied at different scales, aggregating information across subgraphs allows for uncovering multi-scale structure (coarsening). A number of classical algorithms implement such operations, including spectral clustering (22; 51), the Louvain algorithm (11) and Graclus (14). Recently, clustering approaches based on graph curvature (41; 50; 59; 54; 19) have received interest. They utilize discrete Ricci curvature (43; 23) to characterize the local geometry of a graph by aggregating geometric information across 2-hop node neighborhoods. An associated curvature flow (42) characterizes graph geometry at a more global scale, uncovering its coarse geometry.

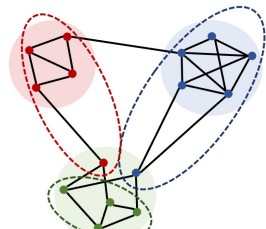

Figure 1: Clustering based on connectivity vs. node attributes only.

A crucial shortcoming of classical node clustering approaches in the case of *attributed graphs* is their inability to capture structural information encoded in the node attributes, as they evaluate only similarity structure encoded in the graph's topology. However, such information is often vital for downstream tasks. Applying a

separate clustering approach to the (typically vector-valued) node attributes does not resolve this issue, as it is blind to similarity structure encoded in the graph's topology, which is often equally vital for downstream tasks (Fig. 1). Hence, to capture meaningful clusters in attributed graphs, pooling operators need to evaluate *both* graph topology and node attributes. Recent literature has introduced a plethora of pooling operators for attributed graphs (9; 65; 4), many of which are based on the classical algorithms listed above. In this work, we extend, for the first time, clustering approaches based on Ricci flow to attributed graphs.

**Related Work.** The design of pooling layers has historically build on clustering algorithms, such as GRACLUS (14) or MINCUTPOOL (8). A plethora of pooling operators has been proposed in recent literature, inspired by spectral clustering (9; 32), matrix factorization (4), hierarchical clustering (65), modularity (39) and multi-set encoding (5), among others. The *Select-Reduce-Connect* (short: *SRC*) *framework* (28) serves as a unifying framework and benchmark for pooling operators. Most closely related to our work is a pooling approach proposed by (47), which performs edge cuts guided by Ricci curvature, instead of a curvature-adjustment computed via Ricci flow. Among other differences, their approach does not incorporate node attributes, which can lead to reduced performance on attributed graphs. We provide a detailed comparison of the two approaches (conceptual and experimental) in Apx. C.

**Summary of contributions.** The main contributions of the paper are as follows:

1. We introduce ORC-POOL, a trainable *pooling operator*, which utilizes discrete Ricci curvature and an associated geometric flow to identify salient multi-scale structure in graphs. ORC-POOL groups nodes according to similarity structure encoded *both* in the graph's topology, as well as its nodes' attributes, presenting the first extension of Ricci flow based clustering to attributed graphs.

2. We further introduce a pooling layer, which allows for incorporating ORC-POOL into Message-Passing Graph Neural Networks.

3. We perform a range of computational experiments, which demonstrate the utility of ORC-POOL layers through improvements in node- and graph-level tasks. We complement our empirical results with a discussion of the structural properties of ORC-POOL.

## 2 Background and Notation

Let $G = \{V, E\}$ denote a graph, $V$ ($|V| = N$) the set of nodes and $E = V \times V$ the set of edges; $G$ is endowed with the usual shortest-path metric $d_G$. We assume that $G$ is attributed and denote node attributes as $X \in \mathbb{R}^{|V| \times m}$. We further assume that $G$ is *weighted*; i.e., its edges are endowed with scalar attributes, given by a weight function $w : E \to \mathbb{R}_+$. Below, we recall basic Graph Neural Networks concepts and introduce the graph curvature notions and associated curvature flow utilized in this work.

### 2.1 Graph Neural Networks with Pooling

**Message-passing Graph Neural Networks.** The blueprint of many state of the art architectures are Message-Passing GNNs (MPGNNs) (25; 30), which learn node embeddings via an iterative "message-passing" (MP) scheme: Node representations are iteratively updated as a function of their neighbors' representations. Node attributes in the input graph determine the node representations at initialization. Formally, MPGNNs consist of a *message* $m_v^{(l+1)}$ and an update function $f_{Up}$ ($l = 0, \ldots, L-1$ denoting the layer):

$$m_v^{(l+1)} = f_{Agg}^{(l)}\big(h_v^{(l)}, \{h_u^{(l)} \mid u \in \mathcal{N}_v\}\big)$$
$$h_v^{(l+1)} = f_{Up}\big(h_v^{(l)}, m_v^{(l+1)}\big) \,.$$

$f_{Agg}, f_{Up}$ may be implemented via MLPs with trainable parameters. Popular examples of MPGNNs are GCN (34), GIN (63), GraphSage (30) and GAT (56).

**Graph pooling operators.** State of the art GNN architectures often combine MP base layers with pooling layers. The *selection, reduction, and connection* (short: *SRC*) *framework* (28) formalizes *pooling operators* as

maps POOL: $G \mapsto G^P = (X^P, E^P)$ composed of three functions that act on the nodes, node attributes and edges of $G$, respectively (Fig. 3):

1. A *selection* function, which identifies a set of nodes that are merged to supernodes ($\mathcal{V}_k \subseteq \{1, \ldots, N\}$): SEL : $\{1, \ldots, N\} \mapsto (\mathcal{V}_1, \ldots, \mathcal{V}_K)$. The selection function lies at the heart of the pooling operator. Below, we propose a *geometric selection function*, which utilizes the graph's geometry for node-level clustering.

2. A *reduction* function, which computes the attributes of supernodes $\mathcal{V}_k$ by aggregating the attributes of its nodes: RED : $\{(x_{k_1}, x_{k_2}, \ldots)\}_{k \in [K]} \mapsto \{x'_k\}_{k \in [K]}$.

3. A *connection* function, which determines the connectivity of supernodes and (re-)assigns edge attributes: CON : $((\mathcal{V}_1, \ldots, \mathcal{V}_K), E) \mapsto \{(\mathcal{V}_k, \mathcal{V}_l), e'_{kl}\}_{k,l \in [K]}$.

SRC is a unifying framework for commonly used pooling operators and has been utilized in graph machine learning libraries such as SPEKTRAL(27) and PYTORCH GEOMETRIC(21).

## 2.2 Graph Curvature

Classically, Ricci curvature establishes a connection between the local dispersion of geodesics and the local curvature of a manifold, deriving from a crucial connection between volume growth rates and curvature: Negative curvature characterizes exponential fast volume growth, while positive curvature indicates contraction. Several analogous notions have been proposed for discrete spaces, including by Ollivier (43), Forman (23) and Erbar-Maas (18). Common among all notions is the observation that edges, which encode long-range dependencies, have low (negative) curvature, while edges that form local connections have high curvature, allowing for a characterization of local and global connectivity patterns in a graph. Here, we focus on Ollivier's curvature, which we introduce below.

### 2.2.1 Ollivier's Ricci curvature

Ollivier (43) introduces a Ricci curvature, which relates the curvature along a geodesic between nearby points $x, y$ on a manifold $\mathcal{M}$ with the transportation distance between their neighborhoods. Recall that the Wasserstein-1 distance between two probability distributions $\mu_1, \mu_2$ is given by

$$W_1(\mu_1, \mu_2) = \inf_{\mu \in \Gamma(\mu_1, \mu_2)} \int d_{\mathcal{M}}(x, y) \mu(x, y) \, dx \, dy \, , \tag{1}$$

where $d_{\mathcal{M}}(x, y)$ denotes the geodesic distance, $B_{\mathcal{M}}(x, \epsilon) := \{z \in \mathcal{M} : d_{\mathcal{M}}(x, z) \leq \epsilon\}$ the $\epsilon$-neighborhood of $x$ and $\Gamma(\mu_1, \mu_2)$ the set of measures with marginals $\mu_1, \mu_2$. Further let $\mu_x^{\mathcal{M}}(z)$ denote a measure on this neighborhood. Then *Ollivier's Ricci curvature* (ORC) is given by $\kappa_{\mathcal{M}}(x, y) = 1 - \frac{W(\mu_x^{\mathcal{M}}, \mu_y^{\mathcal{M}})}{d_{\mathcal{M}}(x,y)}$. Analogous to the continuous case, ORC can be defined in discrete spaces: Let $B$ denote a matrix with entries $b_{ij} = e^{-w_{ij}}$ inverse proportional to the edge weights and $D_B$ denote a matrix with entries $d_{ii} = \sum_j b_{ij}$. Consider a diffusion process on the graph $G$, i.e., we place a point mass $\delta_v$ at a node $v$ and consider the diffusion $p_v(t) = \delta_v e^{-tD_B^{-1}(D_B - B)}$ (12). The distribution generated by $p_v(t)$ defines a measure on the $t$-neighborhood of $v$. We define a discrete analog of ORC (43; 26) for an edge $e = (u, v)$ via the transportation distances between neighborhood measures defined by diffusion processes starting at its adjacent vertices, i.e.,

$$\kappa_{uv}^t = 1 - \frac{W_1(p_v(t), p_u(t))}{d_G(u, v)} \, . \tag{2}$$

How does this quantity relate to the coarse structure of the graph? In most graphs, similar nodes form densely connected subgraphs (*homophily principle*). As a result, if $u, v$ are similar, then the diffusion processes will stay nearby with high probability, exploring the densely connected subgraph. In contrast, if $u, v$ are dissimilar, e.g., belong to separate clusters, then the diffusion processes are likely to explore separate clusters, drawing apart quickly. The transportation distance $W_1(p_v(t), p_u(t))$ is much higher in the second case than in the first. Thus, edges that connect dissimilar nodes have low (usually negative) curvature, whereas edges that connect similar nodes have high curvature. Hence, bridges between communities may be identified via low curvature

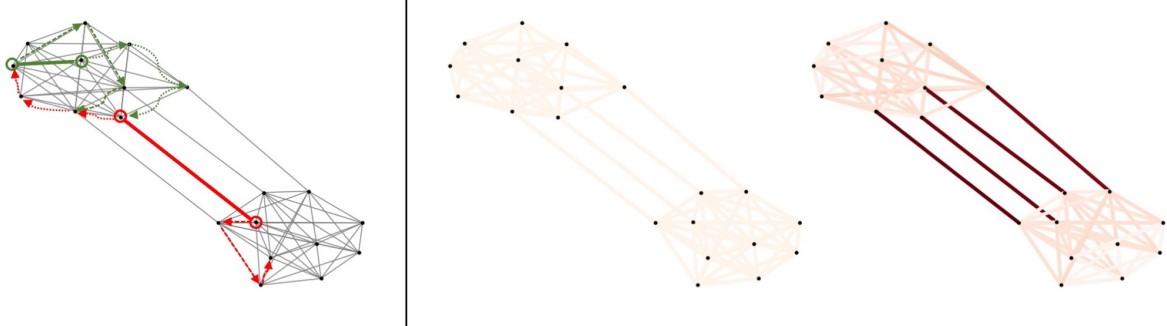

Figure 2: Discrete Ricci curvature reveals coarse structure. **Left:** Relation between curvature and trajectories of diffusion processes starting at similar (green) and dissimilar (red) nodes. **Right:** Dumbbell graph with uniform edge weights (at initialization) and with curvature-adjusted edge weights (darker colors represent lower curvature).

(see Fig. 2). To ensure computational feasibility, we restrict ourselves to a first-order approximation of the diffusion process, $p_v \approx \alpha I + (1-\alpha)\delta_v D_B^{-1} B$, which defines a measure on the 1-hop node neighborhoods. With this choice, we recover the classical discrete Ollivier-Ricci curvature (43; 36). We note that the pooling operations introduced below naturally extend to curvature computed over $t$-hop neighborhoods. Varying the neighborhood radius could provide an additional avenue for incorporating multi-scale structure into the curvature computation, which could be valuable in practise, albeit at a higher computational cost. Notice that the curvature notion introduced above does not account for node attributes. We may define edge weights that encode a similarity measure on the node attributes, e.g., $w_{ij} = \frac{1}{m+1} \sum_k \mathbf{1}_{\{x_i^k \neq x_j^k\}}$).

### 2.2.2 Curvature-based Clustering

**Ricci Flow.** In continuous spaces, an associated geometric flow (*Ricci flow*) is of great importance, as it reveals deep connections between the geometry and topology of the manifold (geometrization conjecture). As a loose analogy, a geometric flow associated with discrete Ricci curvature can be defined, which has been related to the community structure (41) and coarse geometry (61) of graphs. Ollivier (42) proposes a curvature flow

$$\frac{d}{dt}d_G(u,v)(t) = -\kappa_{uv}(t) \cdot d_G(u,v)(t) \quad ((u,v) \in E) ,$$

associated with discrete ORC along edges $(u,v)$. A discretization of this flow gives a combinatorial evolution equation, which evolves edge weights according to the local geometry of the graph: Consider a family of weighted graphs $G^t = \{V, E, W^t\}$, which is constructed from an input graph $G = G^0$ by evolving its edge weights as (setting $dt = 1$)

$$w_{u,v}^t \leftarrow (1 - \kappa_{uv})d_G(u,v) \qquad ((u,v) \in E) , \tag{3}$$

where $\kappa_{uv}, d_G(u,v)$ are computed on $G^t$. Eq. 3 may be viewed as a discrete analogue of continuous Ricci flow. To control the scale of the edge weights, we normalize after each iteration.

**Geometric clustering algorithms.** Geometric clustering approaches based on ORC for non-attributed graphs have previously been considered, e.g. (50; 41; 54). These algorithms are partition-based, i.e., remove edges to decompose the graph into subgraphs, and come in two flavors: The first takes a local perspective, cutting edges with ORC below a given threshold (e.g., (50)). The second evolves edge weights under Ricci flow (Eq. 3), encoding local and more global geometric information into the edge weights, before removing edges with weight below a certain threshold (e.g., (41)). Both approaches exploit the observation that ORC flow "highlights" the multi-scale structure of the graph (e.g., community structure). We emphasize that curvature, by construction, evaluates only the graph's connectivity, but cannot account for node attributes (apart from carefully chosen initialization of edge weights, see above).

### 2.2.3 Curvature Approximation

Computing ORC as defined above involves solving an optimal transport problem (i.e., computing the Wasserstein-1 distances between measures on the 1-hop neighborhoods of the adjacent vertices). In the discrete setting, this corresponds to computing the earth mover's distance, which, in the worst case, has complexity $O(|E|m^3)$ (where $m$ denotes the maximum degree of nodes in $G$). A commonly used approximation via Sinkhorn's algorithm has a reduced complexity of $O(|E|m^2)$, which can still be prohibitively expensive on large-scale graphs. Recently, Tian et al. (54) introduced a combinatorial approximation of ORC of the form $\widehat{\kappa}_{uv} := \frac{1}{2}\left(\kappa_{uv}^{up} + \kappa_{uv}^{low}\right)$, where $\kappa^{up}$ and $\kappa^{low}$ are combinatorial upper and lower bounds, which can be computed in $O(|E|m)$. More details, as well as the formal statement of the bounds can be found in Apx. A. In large or dense graphs, implementing the above introduced Ricci flow via this approximation can significantly reduce runtime (54).

## 3 Geometric Coarsening and Pooling in attributed graphs

Characterizing the coarse geometry of a graph is invaluable for graph learning tasks. In this paper, we introduce a geometric pooling operator (ORC-POOL, Fig. 3), which may serve as a standalone graph coarsening approach, as well as a pooling layer that can be integrated into GNN architectures. ORC-POOL allows for evaluating similarity structure encoded in the graph's geometry via ORC flow, while also accounting for similarity structure encoded in the node attributes. We follow the SRC framework (28) discussed above to formalize our approach.

### 3.1 Geometric Graph Coarsening

**Geometric selection function.** To define a pooling operator that effectively coarsens a graph, the selection function needs to identify a cluster assignment that preserves the overall graph topology (community structure, etc.). We propose a *geometric selection function*, which computes the cluster assignment guided by the relation between curvature and graph topology (sec. 2.2). We encode curvature information into a matrix $C_T = (c_{ij})$ with entries $c_{ij} = w_{ij}$ given by the evolved edge weights after $T$ iterations of ORC flow. The matrix $C$ can be seen as a *curvature-adjusted* adjacency matrix, which assigns an *importance score* to each edge reflecting its structural role: Under ORC flow (Eq. 3), structurally important edges are assigned a high score, whereas edges between similar nodes receive a low score. This encodes global similarity structure, uncovering the coarse geometry of the graph (see discussion above and Fig. 2). Edges whose weight decreases under ORC flow contract (positive curvature), moving the adjacent, similar nodes closer together. On the other hand, edges whose weight increases under ORC flow expand (negative curvature), moving the adjacent dissimilar nodes further apart. This naturally induces a coarsening of the graph. Curvature-based coarsening utilizes this emerging structure by cutting edges with a high weight ($w_{ij}^T > \Delta$) and then merging the remaining connected components into supernodes. The threshold $\Delta$ is a crucial hyperparameter, which is often difficult to choose in practice. Ni et al. (41) learn the threshold by optimizing the modularity of the resulting decomposition. Modularity optimization is NP-hard and hence an exact solution is intractable. They approximate the solution via an expensive parameter search. Since this subroutine is both expensive and non-differentiable, it is not suitable for integration into a trainable architecture. Hence, we need to employ a different auxiliary loss for identifying edges to cut, which we now describe.

**Graph Cuts.** Let $S \in \{0,1\}^{N \times K}$ denote the assignment matrix computed by the selection function, its entries specify the assignment of $N$ vertices to $K$ supernodes, i.e., $s_{ik} = 1$, if vertex $i$ is $\mathcal{V}_k$ and $s_{ik} = 0$ otherwise. The problem of computing $S$ by removing edges relates to the *mincut problem*, a classical problem in Combinatorics. In a seminal paper, Shi and Malik (49) show that the min-cut problem can be written as an optimization problem with respect to the number of edges within and between putative clusters (here, supernodes):

$$\max \frac{1}{K}\sum_{k=1}^{K} \frac{\sum_{i,j \in V_k} \mathbf{1}_{i \sim j}}{\sum_{i \in V_k, j \in V \setminus V_k} \mathbf{1}_{i \sim j}} \ . \tag{4}$$

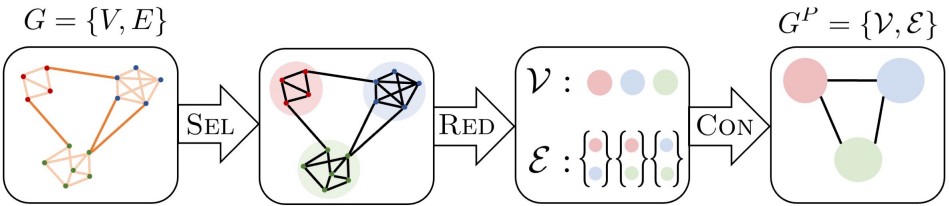

Figure 3: Proposed geometric pooling operator (ORC-POOL), which utilizes a curvature-based, geometric selection function (SEL) to identify supernodes and superedges (RED), which are then reconnected to generate the pooled graph (CON).

Here, $\mathbf{1}_{i \sim j}$ denotes the edge indicator function. In the Graph ML literature, an equivalent formulation of 4 due to Dhillon et al. (13) has been widely adapted,

$$\max \frac{1}{K} \sum_{k=1}^{K} \frac{S_k^T A S_k}{S_k^T D S_k} \qquad \text{s.t. } S\mathbf{1}_K = \mathbf{1}_N , \tag{5}$$

where $A$ denotes the graph's (unweighted) adjacency matrix, $D = \text{diag}(A)$ its degree matrix and $S_k$ the $k$th row of $S$. While this problem itself is NP-hard, it can be efficiently solved via the relaxation (52)

$$Q^* := \underset{\substack{Q \in \mathbb{R}^{N \times K} \\ Q^T Q = I_K}}{\text{argmax}} \frac{1}{K} \sum_{k=1}^{K} Q_k^T A Q_k \qquad \text{s.t.} \quad Q = D^{1/2} S (S^T D S)^{-1/2} . \tag{6}$$

Using classical results from spectral graph theory, one can show that the optimizer takes the form $Q^* = U_K V$, where $U_K \in \mathbb{R}^{N \times K}$ is formed by the top $K$ eigenvectors of the normalized adjacency $\hat{A} = D^{-1/2} A D^{-1/2}$ and $V \in O(K)$ is an orthogonal transform. To recover the cluster assignment $S$, one can apply $k$-means clustering to the rows of $Q^*$ (57). Approaches for efficiently computing minimal graph cuts have been adapted for the design of effective graph pooling operators (9; 31). We have argued above that computing a curvature-adjustment emphasizes the underlying community structure (e.g., Fig 2), which should make it easier to learn graph cuts. Note that after curvature-adjustment, the graph is weighted, i.e., we now optimize the sum of edge weights within and between putative clusters. We will provide theoretical evidence for this observation in the next section. Weighted graph cuts have been considered previously, e.g., in METIS (14), which underlies the GRACLUS pooling layer. We can adapt Eq. 6 to a loss function for computing a cluster assignment based on the (normalized) curvature-adjusted adjacency matrix, i.e.,

$$\min_S \left[ \mathcal{L}_{\text{CP}} = -\frac{\text{tr} S^T \hat{C}_T S}{\text{tr} S^T \hat{D} S} \right] . \tag{7}$$

Note that the entries of the degree matrix $\hat{D}$ are the weighted node degrees after adjustment. For $T = 0$, we recover the classical minimum graph cut objective. Notably, the resulting objective is differentiable and can be optimized with standard techniques (see above), which is preferable over the parameter search for a suitable edge weight threshold performed in (50; 41).

**Geometric pooling operator (ORC-Pool).** The reduction and connection functions of ORC-POOL closely resemble canonical choices that are common among many pooling operators (e.g., MIN-CUTPOOL (8), DIFFPOOL (65), NMF (4), among others): Given a selection $S \in \{0,1\}^{N \times K}$, which assigns $N$ nodes to $K$ supernodes, we set $\text{RED}(X) = S^T X =: X'$ and $\text{CON}(E) = S^T E S := E'$. Naturally, if $G$ is not attributed, then there is no need for a reduction function. In the coarsened graph, edge weights are again initialized either as 1 or according to differences in supernode attributes (implemented in the CON function). This coarsening scheme may be applied multiple times, i.e., we can learn a sequence $S_1, S_2, \ldots$ of selection matrices, which compute cluster assignments with varying degree of coarseness. With that, ORC-POOL may serve as a standalone coarsening or clustering approach.

### 3.2 ORC-Pool layer for Graph Neural Networks

While the geometric pooling operator described above applies to attributed graphs, its selection function does not evaluate similarity structure encoded in node attributes, aside from a specific edge weight initialization discussed earlier. In this section, we will describe an alternative coarsening approach, where ORC-POOL is integrated into GNNs as a pooling layer grounded in graph geometry. Stacked on top of MP base layers, which learn node representation that reflect similarity structure in both graph topology and node attributes, ORC-POOL integrates both types of structural information to coarsen the graph.

Pooling operators can be integrated into GNNs by stacking pooling layers, which implement the operator, on top of blocks of base layers (usually MP layers). The base layer learns node embeddings $\tilde{X} = \text{GNN}(X, \hat{A}, \theta)$, where $\theta$ are trainable parameters. If the input graph is attributed, its node attributes are used to initialize the base layer. We implement the ORC-POOL layer as an MLP, where we learn the cluster assignment matrix $S$ with an MLP with softmax activation (enforcing the constraint $S\mathbf{1}_k = \mathbf{1}_N$) and trainable parameters $\psi$ ($S = MLP(\tilde{X}, \psi)$), which are trained using Eq. 7 as an auxiliary loss. The GNN is trained end-to-end. In particular, we optimize the training objective

$$\min \left[ \mathcal{L}_{\text{CP}}(\theta, \psi) = -\frac{\text{tr } S^T \hat{C}_T S}{\text{tr } S^T \hat{D} S} + \left\| \frac{S^T S}{\|S^T S\|_F} - \frac{I_K}{\sqrt{K}} \right\|_F \right] , \tag{8}$$

where the first term of the objective corresponds to the auxiliary loss and the second encourages mutually orthogonal clusters of similar size, preventing degenerate clusters ($\|\cdot\|_F$ denotes the Frobenius norm). Notice that the training objective closely resembles that of MINCUTPOOL (8), but utilizes the curvature-adjusted adjacency matrix $C_T$. In particular, we recover MINCUTPOOL as a special case ($T = 0$).

We obtain the adjacency matrix of the coarsened graph by setting $A^P = S^T A S$ and recomputing curvature-adjusted weights via Ricci flow on the superedges. We initialize superedge weights as a measure of similarity of the supernode attributes, which are obtained by setting $X^P = S^T X$. Stacking blocks of base layers and pooling layers on top of each other yields a successive coarsening of the graph, which reflects its multi-scale structure. Notice that curvature-based importance scores influence the message functions on each scale. The trainable parameters $\psi, \theta$ are trained end-to-end as part of the GNN architecture and the respective loss for the downstream task, allowing for a seamless integration of geometric pooling into GNN architectures. As a final remark, notice that the curvature-adjustment employed in ORC-POOL is flexible and could be incorporated into other (dense) pooling operators too (especially those that build on graph cuts).

## 4 Properties of ORC-Pool

### 4.1 Basic properties

**Permutation-invariance.** Graphs and sets are permutation-invariant, as there is no natural order relation on the nodes. Standard MP layers are permutation-equivariant and pooling layers permutation-invariant to encode this structure as inductive bias. The curvature-adjustment added in ORC-POOL preserves this property (for details see Apx. G).

**Impact on Expressivity.** A classical measure of the representational power of GNNs is *expressivity*, which evaluates a GNNs ability to distinguish pairs of non-isomorphic graphs. It is well-known that standard MPGNNs are as powerful as the Weiserfeiler-Lehmann test, a classical heuristic for graph isomorphism testing (63; 38). To reap the benefits of pooling, pooling operators should preserve the expressivity of the MP base layer. Recently, (10) established conditions for this property. A simple corollary shows that ORC-POOL fulfills these conditions (see Apx. G for details):

**Corollary 1.** *Consider a simple architecture with a block of MP base layers, followed by a ORC-POOL layer. Let $G_1, G_2$ denote two 1-WL-distinguishable graphs with node attributes $X_1, X_2$. Further let $X_1' \neq X_2'$ denote the node representations learned by the block of MP layers. Then the coarsened graphs $G_1^P, G_2^P$ learned by the ORC-POOL layer are 1-WL distinguishable.*

**Topological Effects.** MPGNNs are known to suffer from *oversmoothing* (35), which describes the convergence of the representations of dissimilar nodes in densely connected subgraphs as the number of layers increases. This effect is particularly prevalent in node-level tasks, e.g., negatively impacting the GNN's ability to perform node clustering. ORC-POOL mitigates this effect in two ways: By assigning higher weights to sparse connections between dense subgraphs and smaller weights to edges within dense subgraphs, distances between similar nodes are contracted and distances between dissimilar nodes are expanded. Moreover, pooling layers generally counteract oversmoothing, as they induce scale-separation. That is, local and global features are preserved separately, as they are encoded on different coarsening scales. Since similar nodes are merged into supernodes, neighboring supernodes tend to be less similar than neighboring nodes at the previous scale, alleviating oversmoothing on coarser scales. Conversely, limiting the number of layers could avoid over-smoothing, but shallow MPGNNs are known to suffer from under-reaching (7): If the number of layers is smaller than the graph's diameter, information between distant nodes is not exchanged, which can limit the utility of the learned representations in downstream tasks. This underscores the value of effective pooling layers for the design of deeper GNNs. We comment on the impact of pooling on over-squashing, a related effect, in Apx G.

## 4.2 Structural Impact of Curvature Adjustment

We further corroborate our earlier claim that ORC flow reveals coarse geometry and emphasizes the community structure, making it easier to learn suitable graph cuts. For our analysis, we employ a graph model, which exhibits community structure as found in real data:

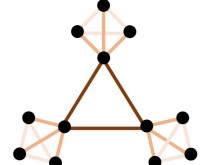

Figure 4: $G_{a,b}$ $(a = b = 3)$

**Definition 1.** *Consider a class of model graphs $G_{a,b}$ $(a \geq b \geq 2)$, which are constructed by taking a complete graph with $b$ nodes and replacing each by a complete graph with $a + 1$ nodes.*

$G_{a,b}$-graphs can be seen as instances of stochastic block models with $b$ blocks of size $a + 1$ and exhibit a clear community structure (see Fig. 9 for an example). Due to the regularity of the graph, we can categorize its edges into three types: (1) *bridges*, which connect dissimilar nodes in different clusters; (2) *internal* edges, which connect similar nodes in the same cluster that are at most one hop removed from a dissimilar node; and (3) all other *internal* edges, which connect similar nodes within the same cluster. (41) derived an evolution equation for edge weights under ORC flow for $G_{a,b}$ graphs:

**Lemma 1** (informal, (41)). *Let $w^t = [w_1^t, w_2^t, w_3^t]$ denote the weights of edges of types (1)-(3). Assuming $w^0 = [1, 1, 1]$, the edge weights evolve under ORC flow (Eq. 3) as $w^{t+1} = F(a,b)w^t$, where $F(a,b)$ is a fixed $(3 \times 3)$-matrix, depending on $a, b$ only.*

Utilizing this result we want to analyze the strength of the community structure subject to Ricci flow, employing modularity (24) as a measure:

$$Q(W) := \frac{1}{\sum_{uv} w_{uv}} \sum_{uv} \left( w_{uv} - \frac{d_u d_v}{\sum_{uv} w_{uv}} \right) \delta(C_u, C_v) .$$

Here, $W = (w_{ij})$ denotes a weighted adjacency matrix, $d_v$ weighted node degrees and $\delta(C_u, C_v) = 1$, if $u, v$ are in the same cluster and zero otherwise. High modularity is often used as an indicator of a "strong" separation of the graph into subgraphs, under which communities can be easier to detect. We show that modularity increases as edge weights evolve under ORC flow:

**Theorem 1** (informal). *Let $W$ denote the original adjacency matrix and $C_t$ the curvature-adjusted adjacency matrix after $t$ iterations of ORC flow. Then $Q(C_t) > Q(W)$, with $Q(C_t)$ increasing in $t$.*

We defer the full technical statement of Lem. 3 and Thm. 5, as well as proof details to supp. G.

Table 1: **Node Clustering.** Average accuracy (NMI) for ORC-POOL in comparison with state of the art pooling layers; average time per epoch (in seconds) is given in brackets. The reported times (mean/ standard deviation) are computed based on 10 trials. Highest accuracy in bold.

| Layer | Cora | | CiteSeer | | PubMed | |
|---|---|---|---|---|---|---|
| No pool | $0.33 \pm 0.04$ | $(0.008 \pm 0.000)$ | $0.22 \pm 0.02$ | $(0.009 \pm 0.000)$ | $0.14 \pm 0.02$ | $(0.060 \pm 0.001)$ |
| Diff | $0.20 \pm 0.04$ | $(0.010 \pm 0.000)$ | $0.24 \pm 0.12$ | $(0.009 \pm 0.000)$ | $0.03 \pm 0.02$ | $(0.082 \pm 0.002)$ |
| Mincut | $0.43 \pm 0.03$ | $(0.018 \pm 0.001)$ | $0.31 \pm 0.04$ | $(0.016 \pm 0.001)$ | $0.23 \pm 0.04$ | $(0.689 \pm 0.003)$ |
| DMoN | $0.37 \pm 0.05$ | $(0.010 \pm 0.000)$ | $0.29 \pm 0.03$ | $(0.010 \pm 0.000)$ | $0.18 \pm 0.04$ | $(0.064 \pm 0.001)$ |
| TV | $0.33 \pm 0.05$ | $(0.010 \pm 0.000)$ | $0.30 \pm 0.05$ | $(0.010 \pm 0.000)$ | $0.21 \pm 0.03$ | $(0.069 \pm 0.001)$ |
| Graclus | $0.43 \pm 0.03$ | $(0.012 \pm 0.001)$ | $0.34 \pm 0.02$ | $(0.013 \pm 0.001)$ | $\mathbf{0.27 \pm 0.02}$ | $(0.138 \pm 0.001)$ |
| ORC (us) | $\mathbf{0.47 \pm 0.04}$ | $(0.035 \pm 0.000)$ | $\mathbf{0.35 \pm 0.04}$ | $(0.029 \pm 0.001)$ | $0.24 \pm 0.04$ | $(0.92 \pm 0.003)$ |

Table 2: **Graph Classification.** Average classification accuracy for ORC-POOL in comparison with state of the art pooling layers, averaged over 10 trials, using a 80/10/10 train/val/test split. Highest accuracy in bold.

| Layer | MUTAG | ENZYMES | PROTEINS | IMDB-BINARY | REDDIT-BINARY | COLLAB | PEPTIDES |
|---|---|---|---|---|---|---|---|
| No Pool | $0.81 \pm 0.05$ | $0.29 \pm 0.04$ | $0.74 \pm 0.03$ | $0.58 \pm 0.02$ | $0.85 \pm 0.03$ | $0.67 \pm 0.03$ | $0.63 \pm 0.03$ |
| Diff | $0.83 \pm 0.09$ | $0.31 \pm 0.03$ | $0.75 \pm 0.04$ | $0.64 \pm 0.06$ | $0.86 \pm 0.02$ | $0.70 \pm 0.02$ | $0.66 \pm 0.02$ |
| Mincut | $0.83 \pm 0.07$ | $0.37 \pm 0.05$ | $0.76 \pm 0.05$ | $0.65 \pm 0.05$ | $0.85 \pm 0.02$ | $0.67 \pm 0.02$ | $0.67 \pm 0.02$ |
| DMoN | $0.84 \pm 0.08$ | $0.40 \pm 0.04$ | $0.76 \pm 0.05$ | $0.65 \pm 0.04$ | $0.85 \pm 0.02$ | $0.68 \pm 0.02$ | $0.67 \pm 0.02$ |
| TV | $0.83 \pm 0.08$ | $0.37 \pm 0.05$ | $0.75 \pm 0.04$ | $0.54 \pm 0.03$ | $0.85 \pm 0.03$ | $0.70 \pm 0.02$ | $0.67 \pm 0.02$ |
| Graclus | $0.84 \pm 0.06$ | $\mathbf{0.42 \pm 0.03}$ | $0.75 \pm 0.04$ | $0.61 \pm 0.04$ | $0.84 \pm 0.04$ | $0.68 \pm 0.02$ | $0.66 \pm 0.03$ |
| ORC (us) | $\mathbf{0.90 \pm 0.06}$ | $0.38 \pm 0.06$ | $\mathbf{0.78 \pm 0.04}$ | $\mathbf{0.71 \pm 0.04}$ | $\mathbf{0.88 \pm 0.02}$ | $\mathbf{0.71 \pm 0.02}$ | $\mathbf{0.69 \pm 0.01}$ |

## 5 Experimental Analysis of Geometric Pooling

In this section we present experiments to demonstrate the advantage of our proposed pooling layer ORC-POOL. We test our **hypothesis** that *encoding local and global geometric information into the pooling layers can increase the accuracy of the GNN in downstream tasks.*

**Experimental setup.** We implement a simple GNN architecture, consisting of blocks of GCN base layers, followed by a pooling layer. When comparing ORC-POOL with other state of the art pooling layers, we keep the architecture fixed (only altering pooling layers) to enable a fair comparison. Details on the architecture used in our node- and graph-level experiments can be found in Apx. D.1. We utilize the popular benchmarks PLANETOID (64) for node clustering, and TUDATASET (37) and LRGBDATASET (17) for graph classification. Experiments are performed using PYTORCH GEOMETRIC. We compare ORC-POOL with four state of the art pooling layers, which were reported as best-performing across domains and graph learning tasks (28): MINCUTPOOL (8), DIFFPOOL (65), TVPOOL (32), and Deep Modularity Networks (DMON) (55). We implement ORC-POOL using exact ORC, i.e., $W_1(\cdot, \cdot)$ is computed via the earth mover's distance.

**Node Clustering.** We compare the performance of ORC-POOL and other pooling layers for node clustering, where we evaluate the *Normalized Mutual Information* (short *NMI*, defined in sec. D.2) of the cluster assignments computed by the GNN, as well as the average runtime per epoch. The number of desired clusters is known to the model beforehand. Results for PLANETOID are reported in Tab. F. We see that ORC-POOL performed best overall with the highest average NMI on all three graphs. The second best model in all cases was MINCUTPOOL, which the ORC-POOL layer is based on. In terms of runtime, we observed that for the larger PubMed graph, DIFFPOOL, DMON, and TVPOOL are about one order of magnitude faster per epoch compared to MINCUTPOOL and ORC-POOL. The difference in time per epoch is much smaller for Cora and CiteSeer.

**Graph classification.** We further compare the performance of ORC-POOL with that of other pooling layers for graph classification, where we report the accuracy of label assignments. ORC-POOL performed best overall on TUDATASET, achieving the best accuracy on all datasets but ENZYMES (see Tab. 5). We further tested ORC-POOL on PEPTIDES, a large-scale graph classification tasks from the long-range graph benchmark LRGB (17). Here, again, ORC-POOL showed superior performance. Run times for all experiments can be found in Tab. D.3.

Table 3: **Curvature computation.** Comparison of accuracy (NMI) and runtime (in brackets) of computing ORC exactly (EMD), via Sinkhorn distances (Sinkhorn), and via combinatorial ORC approximation. Best runtime in bold.

| Layer | Cora | CiteSeer | PubMed |
|---|---|---|---|
| **EMD** | $0.47 \pm 0.04$ ($19.97 \pm 0.35$) | $0.35 \pm 0.04$ ($16.77 \pm 0.21$) | $0.24 \pm 0.04$ ($571.11 \pm 2.94$) |
| **Sinkhorn** | $0.45 \pm 0.03$ ($56.36 \pm 1.43$) | $0.35 \pm 0.03$ ($42.16 \pm 0.71$) | $0.22 \pm 0.05$ ($904.38 \pm 3.08$) |
| **ORC-approx** | $0.45 \pm 0.03$ ($\mathbf{16.88 \pm 0.43}$) | $0.35 \pm 0.03$ ($\mathbf{14.62 \pm 0.10}$) | $0.21 \pm 0.03$ ($\mathbf{548.43 \pm 1.79}$) |

**Curvature computation.** ORC-POOL defines a *dense* pooling layer; i.e., $\mathbb{E}\left[\mathcal{V}_k/|N|\right] = O(N)$. This property is inherited from the MINCUTPOOL objective and preserved by the curvature adjustment. Hence, the complexity of the pooling operator is $O(NK)$ for storage and $O(K(|E| + NK))$ for minimizing the relaxed min-cut loss (9). As noted in sec. 2.2.3, the computation of the curvature-adjustment can be costly on large, dense graphs. We test the performance of the two introduced curvature approximations (Sinkhorn distances, combinatorial ORC) on the accuracy and average runtime per epoch. We focus on node-level tasks, where input graphs are of larger size. Our results (Tab. 5) indicate that computing Sinkhorn distances is actually slower than EMD on large, dense input graphs. This is consistent with previous results for curvature-based community detection on graphs with similar topology (54). In contrast, the combinatorial ORC approximation delivers a faster method, which retains accuracy similar to exact ORC. We note that our implementation can likely be further optimized, which could lead to additional speedups, e.g., via more effective parallelization.

## 6   Discussion

We introduced ORC-POOL, a geometric pooling operator that leverages coarse geometry, characterized by Ollivier-Ricci curvature and an associated geometric flow. ORC-POOL extends a class of Ricci flow based clustering algorithms to attributed graphs and can be incorporated into GNN architectures as a pooling layer. While curvature adjustment leads to improved performance in downstream tasks, it also adds computational overhead, specifically in node-level tasks with large, dense input graphs. While our experiments indicate that the runtime impact is modest in most cases, we show that a combinatorial ORC approximation can reduce runtime impact while retaining accuracy on node-level tasks. Nevertheless, an important direction for future work is the study of alternative curvature notions or approximations (see Apx. A.2), which could result in more scalable implementations of ORC-POOL. In addition, it would be interesting to investigate the impact of ORC-POOL layers in other graph learning tasks, such as graph regression or reconstruction, as well as on a wider range of graph domains. Lastly, it would be interesting to consider extensions to directed input graphs.

## Acknowledgements

The authors thank Yu Tian for sharing code. AF was supported by the Harvard College Research Program and a KURE Fellowship from the Kempner Institute. MW was partially supported by NSF award CBET-2112085.

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

## Table of Contents

## A  Curvature Approximation

### A.1  Combinatorial ORC Approximation

For completeness, we restate the upper and lower bounds on ORC derived by Tian et al. (54) in our notation. We compute the combinatorial ORC approximation defined in sec. 2.2.3 using these bounds.

Recall the definition of ORC given in Eq. 2, and note that we fix $t = 1$ and $\alpha = 0$. Let $N(x)$ be the 1-hop neighborhood of node $x$. For vertices $u, v$ we have $\ell := N(u) \setminus N(v)$, $r := N(v) \setminus N(u)$, and $c := N(u) \cap N(v)$. Mass distributions on node neighborhoods are denoted as $p_u(x)$. We further define the shorthands $L_u := \sum_\ell p_u(\ell)$ and $L_v := \sum_r p_v(r)$. $a_+$ denotes the quantity $\max(0, a)$.

**Theorem 2** (Lower bound ((54), Thm. 4.6))**.**

$$\kappa_{uv}^1 \geq 1 - \sum_\ell \frac{w_{\ell u}}{w_{uv}} p_u(\ell) - \sum_r \frac{w_{rv}}{w_{uv}} p_v(r)$$
$$- \sum_c \left[ \frac{w_{cv}}{x_{uv}} (p_u(c) - p_v(c))_+ + \frac{w_{cu}}{w_{uv}} (p_v(c) - p_u(c))_+ \right]$$
$$- \left| L_u + p_u(u) - p_v(u) - \sum_c (p_v(c) - p_u(c))_+ \right|.$$

To state the upper bound, we further define $\mathcal{P} := \{x \in N(u) \cup N(y) : p_u(x) - p_v(x) > 0\}$ and $\mathcal{N} := \{x \in N_{(}u) \cup N(y) : p_u(x) - p_v(x) < 0\}$. For $x \in V$ and $S \subseteq V$, let $d_G(x, S) = \min\{d_G(x, y) : y \in S\}$.

**Theorem 3** (Upper bound ((54), Thm. 4.6))**.**

$$\kappa_{uv}^1 \leq 1 - \max \left\{ \sum_x \frac{d_G(x, \mathcal{N})}{w_{uv}} (p_u(x) - p_v(x))_+, \sum_x \frac{d_G(x, \mathcal{P})}{w_{uv}} (p_v(x) - p_u(x))_+ \right\}.$$

### A.2 Forman's Curvature

Forman's curvature (23) gives a combinatorial notion of Ricci curvature, which aggregates local information in the 2-hop neighborhood of an edge. That is, we consider the endpoints of a certain edge and all edges that share an endpoint with the edge. One formalization of this curvature, which considers the contributions of connectivity and triangles for the curvature computation, resembles Ollivier's curvature qualitatively and has found applications in community detection. Its computation is easily parallelizable, allowing for a fast and scalable implementation. This has given rise to a range of applications of Forman's curvature in Network Analysis (60; 53; 48; 58; 62). While in general less suitable for identifying community structure (54), it could present a viable alternative to ORC in large-scale graphs, where computing ORC is infeasible. We can adapt the ORC-POOL operator proposed in this paper to the case of Forman's curvature by simply exchanging the curvature notion $\kappa$.

## B Illustration of Ricci Flow

In the main text, we illustrated the evolution of edge weights under Ricci flow on a dumbbell graph. In Fig. 5, we plot networks sampled from the stochastic block model, with two, three, four and five communities (left to right). We show the network with original weights (all equal to one) on the top and curvature-adjusted edge weights (after 5 iterations of Ricci flow, i.e., $T = 5$) at the bottom, where again darker colors correspond to smaller curvature. We see that curvature reliable uncovers bridges between communities due to their low curvature, whereas edges within communities have a higher weight.

We further illustrate the effect of Ricci flow on molecular graphs, which are representative of graph topologies often encountered in graph-level tasks. Fig. 6 shows the evolution of edge weights under Ricci flow ($T = 4$) for two graphs from the MUTAG data set, again initializing edge weights uniformely with one. We see that the curvature-adjustment highlights "bridges" (dark color) between functional substructures, e.g., rings. This allows the selection function to perform a pooling that aligns with multi-scale structure in the graph.

## C Additional Discussion of Related Work

We provide a more detailed discussion of related work by (47), which also proposes a curvature-based graph pooling approach, albeit with several key differences to our approach. We provide a detailed conceptual comparison of the two approaches below; an experimental comparison can be found in sec. E.3.

**Setting.** Classically, graph pooling layers are designed to jointly evaluate similarity structure encoded in the node features and graph connectivity, instead of connectivity only (see, e.g., DiffPool, MincutPool, as well as

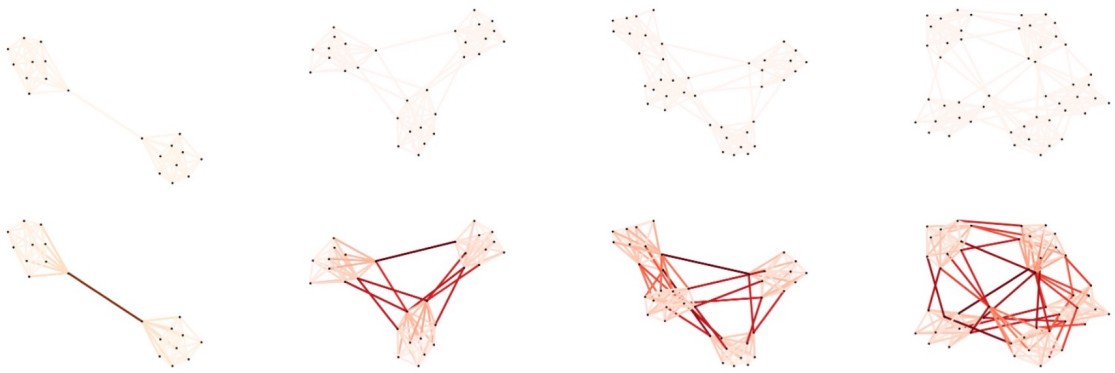

Figure 5: Evolution of edge weights under Ricci flow in the Stochastic Block Model.

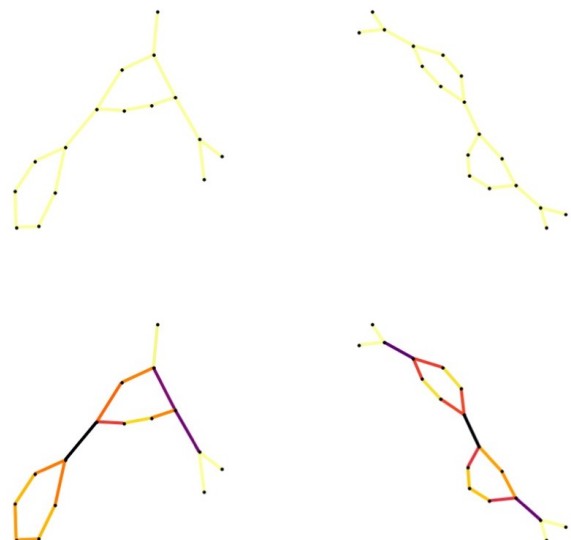

Figure 6: Evolution of edge weights under Ricci flow in MUTAG.

discussions in references such as (28)). Fig. 1 illustrates that in attributed graphs both types of information need to be evaluated to identify meaningful sets of nodes to be pooled. However, the pooling layer introduced in (47) only evaluates similarity structure encoded in the connectivity. This is also corroborated by the experimental comparison of both approaches below, in which ORC-POOL achieves higher performance for all tasks on attributed graphs.

**Motivation.**  The motivation for our proposed pooling layer (and that of related pooling layers) is to capture salient multi-scale structure. In sec. 2.2, we give a geometric motivation for how curvature captures such structure utilizing a connection of discrete Ricci curvature and random walks (see also Fig. 2). On the other hand, the approach in Sanders et al. is motivated by addressing over-smoothing and over-squashing. While we agree that pooling (in general) mitigates over-smoothing due to the induced scale separation (as we discuss in sec. 4.1), over-squashing effects may actually be amplified as we discuss in the appendix.

**Different use of curvature and different notion.**  We note that while both the approach in Sanders et al. and our proposed approach leverages different notions of curvature, there are two fundamental differences: (1) Our approach relies on a curvature-adjusted adjacency matrix, which is computed using Ricci flow, a geometric flow associated with discrete Ricci flow. The approach in Sanders et al. uses a discrete notion of

curvature, it does not use Ricci flow. (2) Our approach utilizes Ollivier's Ricci curvature and approximations thereof. Sanders et al. use a "balanced Forman curvature" instead. While related, the two curvature notions differ substantially. The geometric motivation given in our paper does not directly translate to balanced Forman curvature.

**Complexity and hyperparameters.**  The balanced Forman curvature used in Sanders et al. has complexity $O(|E|d_{max}^2)$. On the other hand, a variant of our proposed approach, which utilizes a combinatorial ORC curvature, has complexity of only $O(|E|d_{max})$, making it more scalable on large-scale graphs. As Sanders et al. state, their approach has several hyperparameters that have to be carefully chosen so as not to simplify the graph too much. Determining this threshold via grid search could add significant computational overhead. On the other hand, the main hyperparameter in our method is the number Ricci flow iterations and the performance is relatively insensitive to the choice of this parameter.

**Overview related pooling methods.**  Tab. 4 gives an overview of our main pooling baselines in the SRC framework.

| Method | Select | Reduce | Connect |
|---|---|---|---|
| DiffPool (65) | $S = \text{GNN}_1(A, X)$ (w/ auxiliary loss) | $X' = S^\top \cdot \text{GNN}_2(A, X)$ | $A' = S^\top A S$ |
| MinCut (8) | $S = \text{MLP}(X)$ (w/ auxiliary loss) | $X' = S^\top X$ | $A' = S^\top A S$ |
| Graclus (15) | $S_k = \{x_i, x_j \mid \arg\max_j \left( \frac{A_{ij}}{D_{ii}} + \frac{A_{ij}}{D_{jj}} \right)\}$ | $X' = S^\top X$ | METIS |
| DMoN (55) | $S = \text{GNN}(\tilde{A}, X)$ (w/ auxiliary loss) | $X' = S^\top X$ | $A' = S^\top A S$ |
| TV (32) | $S = \text{MLP}(X)$ (w/ auxiliary loss) | $X' = S^\top X$ | $A' = S^\top A S$ |

Table 4: Overview of Pooling layers

# D    Additional Details on Experiments

## D.1    Details on Experimental Setup

We implement a GNN architecture consisting of GCN base layers and a single pooling layer, which are jointly trained. For node-level tasks, one GCN layer is used to compute a graph embedding, and another GCN layer is used to compute node clusters from the embedding. Then the pooling layer computes the loss of the current node assignments, which is used to update the GCN parameters. For the graph-level tasks, our architecture alternates between blocks of GCN base layers and pooling layers, which are jointly trained. We train a global pooling layer on top, which computes the readout by aggregating node representations across the graph. To reduce the computational overhead of the ORC computation, we use the curvature-adjustment only in the first pooling layer, i.e., the second pooling layer is a standard MINCUTPOOL layer. We note that the choice of using MINCUTPOOL in subsequent layers was driven by a desire to maximize scalability, as the curvature-adjustment would require recomputing curvature in the coarsened graph. However, we note that computing the curvature-adjustment in subsequent layers has reduced computational cost due to the smaller size of the coarsened graph, which suggests that this is a viable extension of the present approach that may be considered in future work.

To compare against GRACLUS, we incorporated the GRACLUS pooling layer, based on METIS, to PYTORCH GEOMETRIC. We note that our implementation is not optimized for runtime. Hence, we expect that it is possible to obtain further speedups in GRACLUS.

All experiments are run on a NVIDIA A100 GPU with one CPU.

## D.2    Node Clustering

**Architecture.**  The first GCN layer embeds the nodes of the graph into an $m$-dimensional feature space. That is, if the graph initially had a node feature tensor of dimension $n \times k$, where $n$ is the number of nodes, in the output we have a node feature tensor of dimension $n \times m$. The second GCN layer takes this embedded

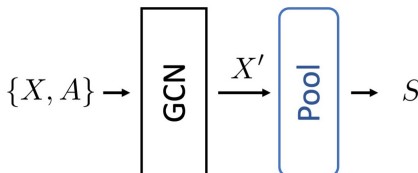

Figure 7: Architecture for node clustering.

graph and computes an assignment tensor of dimension $n \times c$, where $c$ is the number of clusters. By taking the softmax of this $n \times c$ tensor, we obtain the final assignment of each node to a single cluster.

We note that in the original implementation of TVPOOL, custom MP layers are used before pooling. Since the goal of our experiments is a fair comparison of the effectiveness of the pooling layers, we use standard GCN layers as base layers for all pooling approaches.

We also performed the experiments with a GCN without pooling using the same architecture, but omitting the pooling layer. The output is computed by a softmax function, where the predicted class corresponds to the element with the highest value.

**Data.** The PLANETOID graphs Cora, CiteSeer, and PubMed are citation networks where documents are nodes and citations are edges.

Table 5: Planetoid

| Graph | Nodes | Edges | Features | Classes |
|---|---|---|---|---|
| **Cora** | 2708 | 10566 | 1433 | 7 |
| **CiteSeer** | 3327 | 9104 | 3703 | 6 |
| **PubMed** | 19717 | 88648 | 500 | 3 |

**Hyperparameters.** Parameters for the experiments are as follows: For PLANETOID, one GCN layer with output dimension 8 and an ELU activation function is used to embed the graph. The optimizer is an Adam optimizer with a learning rate of $1e-2$. The models are trained for at most 10000 epochs, or until the best NMI is found under a patience constraint. That is, if we achieve the current best NMI on epoch $n$ and patience is $p$, we stop training if the model does not have a better NMI at any epoch up to $n + p$. For the PLANETOID graphs, patience is set to 100, and for Amazon-ratings (from HETEROPHILOUS), patience is set to 250. When applying ORC-POOL to the PLANETOID graphs, four Ricci flow iterations are used, and for Amazon-ratings, two Ricci flow iterations are used.

**NMI.** We measure accuracy as the *Normalized Mutual Information* (short: *NMI*) if the cluster assignments computed by the GNN. NMI is a classical evaluation metric for community detection. In our experiments we employ NMI to measure the accuracy in the node clustering task. We give a brief definition of NMI for completeness.

Let $S \in \mathbb{R}^{N \times k}$ denote a vector that encodes the label assignment to $k$ clusters, i.e., we set $s_{il} = 1$ if node $i$ belongs to cluster $C_l$ and $s_{il} = 0$ otherwise. The entries of $S$ can be viewed as random varibles drawn from the distribution $P(s_l = 1) = N_l/N$ and $P(s_l = 0) = 1 - P(s_l = 1)$, where $N_l := |C_l|$. Using the marginal probability distribution $P_{s_l}$ and the joint probability distribution $P(s_l, s_{l'})$, we further define the entropies $H(s_l)$ and $H(s_l, s_{l'})$, as well as the conditional entropy of $s_l$ given $s_{l'}$ as $H(s_l|s_{l'}) = H(s_l, s_{l'}) - H(s_{l'})$. The NMI for two cluster assignments $S, S'$ is given by

$$NMI(S|S') = 1 - \frac{1}{2} \left( H(S|S') + H(S'|S) \right) .$$

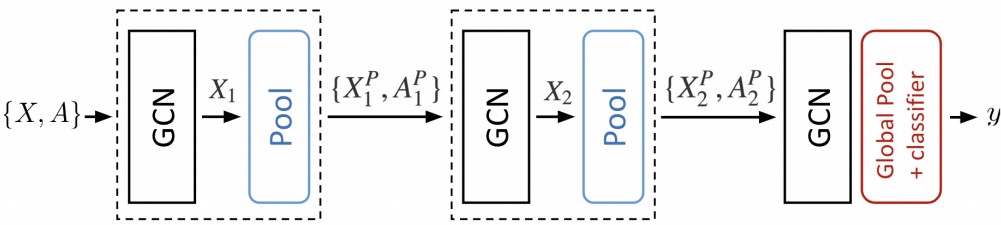

Figure 8: Architecture for graph classification.

### D.3 Graph Classification

**Architecture.** The main part of the model consists of two blocks of GCN layers followed by a pooling layer. The first set of GCN layer(s) embeds the nodes of the graph into an $m$-dimensional feature space. That is, if the graph initially had a node feature tensor of dimension $n \times k$, where $n$ is the number of nodes, in the output we have a node feature tensor of dimension $n \times m$. The second GCN layer takes this embedded graph and computes an assignment tensor of dimension $n \times c$, where $c$ is the number of clusters. By taking the softmax of this $n \times c$ tensor, we obtain the final assignment of each node to a single cluster. Here, the clustering simply gives us a new graph in which groups of nodes have been aggregated, which we pass into the second block. After the second block of GCN layer(s) and pooling, we pass the resulting graph into a third block of GCN layers to get the final node embeddings. These are passed into a global pooling layer, which aggregates node embeddings. In our case, we use global mean pooling, which simply computes the average of all node embeddings. The resulting vector is then passed into a linear classifier.

As an additional baseline, we also run a GCN without pooling on all data sets. The output is obtained using a global mean pooling followed by a linear classifier.

**Data.** Statistics for TUDATASET graphs are shown below (37). MUTAG is a common dataset of chemicals where the task is to predict the mutagenic effect. For ENZYMES, we classify enzymes into one of six classes depending on which type of chemical reaction they catalyze. PROTEINS has a binary classification task for whether a protein is an enzyme. REDDIT-BINARY, COLLAB, and IMDB-BINARY are social networks where we predict the subreddit, research field, and genre of people in the network, respectively. For the graphs that did not have any node features (COLLAB, IMDB-BINARY, REDDIT-BINARY), we add a dummy node feature that is 1.0 for all nodes.

Peptides-func is the only dataset from LRGB asks to classify peptides based on their function.

Table 6: TUDataset and Peptides-func

| Dataset | # of Graphs | Features | Classes |
|---|---|---|---|
| **MUTAG** | 188 | 7 | 2 |
| **ENZYMES** | 600 | 3 | 6 |
| **PROTEINS** | 1113 | 3 | 2 |
| **IMDB-BINARY** | 1000 | 0 | 2 |
| **REDDIT-BINARY** | 1000 | 0 | 2 |
| **COLLAB** | 5000 | 0 | 3 |
| **Peptides-func** | 15535 | 9 | 10 |

**Hyperparameters.** Parameters for the experiments are as follows: For all graphs, the GCN blocks depicted above consist of a GCN layer with output dimension 8 and an ELU activation function. The optimizer is an Adam optimizer with a learning rate of 5e−4 and a weight decay of 1e−4. The models are trained for at most 10000 epochs, or until the best accuracy on a validation is found under a patience constraint. That is, if we achieve the current best validation accuracy on epoch $n$ and patience is $p$, we stop training if the model

does not have a better validation accuracy at any point up to epoch $n + p$. For all datasets, patience is set to 50. When applying ORC-POOL to MUTAG, ENZYMES, and PROTEINS, one iteration of Ricci flow curvature is used. For IMDB-BINARY, REDDIT-BINARY, and COLLAB, two, three, and two iterations are used, respectively. Two iterations are used for Peptides-func. We let the pooling layer approximately halve the number of nodes each time. That is, after the first pooling layer, the number of nodes in $X_1^P$ is the half the average number of nodes for graphs in the dataset. The number of nodes in $X_2^P$ is half the number of nodes in $X_1^P$.

**Runtime comparison.** We report a runtime comparison of the graph-level tasks.

Table 7: Average seconds/epoch on TUDataset and Peptides-func.

|          | MUTAG             | ENZYMES         | PROTEINS          | IMDB-BINARY       |
| -------- | ----------------- | --------------- | ----------------- | ----------------- |
| **No Pool** | $0.023 \pm 0.001$ | $0.13 \pm 0.02$ | $0.22 \pm 0.00$   | $0.16 \pm 0.001$  |
| **Diff**    | $0.057 \pm 0.001$ | $0.17 \pm 0.03$ | $0.34 \pm 0.01$   | $0.29 \pm 0.00$   |
| **Mincut**  | $0.068 \pm 0.001$ | $0.21 \pm 0.03$ | $0.41 \pm 0.00$   | $0.35 \pm 0.01$   |
| **DMoN**    | $0.069 \pm 0.001$ | $0.21 \pm 0.02$ | $0.43 \pm 0.02$   | $0.36 \pm 0.00$   |
| **TV**      | $0.068 \pm 0.000$ | $0.20 \pm 0.02$ | $0.40 \pm 0.01$   | $0.34 \pm 0.02$   |
| **Graclus** | $0.061 \pm 0.002$ | $0.17 \pm 0.01$ | $0.038 \pm 0.01$  | $0.35 \pm 0.02$   |
| **ORC (us)** | $0.190 \pm 0.003$ | $0.48 \pm 0.03$ | $1.32 \pm 0.00$   | $1.07 \pm 0.02$   |

|          | REDDIT-BINARY      | COLLAB          | PEPTIDES-FUNC     |
| -------- | ------------------ | --------------- | ----------------- |
| **No Pool** | $1.68 \pm 0.06$    | $1.02 \pm 0.02$ | $2.85 \pm 0.05$   |
| **Diff**    | $5.28 \pm 0.18$    | $1.74 \pm 0.03$ | $3.79 \pm 0.02$   |
| **Mincut**  | $5.39 \pm 0.17$    | $2.05 \pm 0.01$ | $4.64 \pm 0.07$   |
| **DMoN**    | $2.95 \pm 0.05$    | $2.11 \pm 0.01$ | $4.68 \pm 0.07$   |
| **TV**      | $5.34 \pm 0.09$    | $2.10 \pm 0.02$ | $4.35 \pm 0.05$   |
| **Graclus** | $4.83 \pm 0.11$    | $1.90 \pm 0.03$ | $4.01 \pm 0.04$   |
| **ORC (us)** | $12.90 \pm 0.17$   | $6.05 \pm 0.01$ | $36.27 \pm 0.10$  |

**Results for Heterophilous Graphs.** The HETEROPHILOUS graph Amazon-ratings is a network of Amazon products where edges connect products that are frequently bought together (45). The five classes correspond to product ratings. Unlike the PLANETOID graphs, we do not expect nodes of the same class to consistently cluster together. The "natural" clusters of the graph generally correspond to categories of items, rather than item ratings. Pooling layers combine clusters of nodes in order to coarsen a graph, a procedure whose utility

Table 8: Heterophilous data set.

| Graph             | Nodes | Edges | Features | Classes |
| ----------------- | ----- | ----- | -------- | ------- |
| **Amazon-ratings** | 24492 | 93050 | 300      | 5       |

depends on an implicit homophily assumption. Hence, we do not expect GNNs with pooling layers to perform well on data sets such as the Amazon graph. This is confirmed by our experimental results, which show poor performance (in terms of NMI) for all models. In contrast, in the original HETEROPHILOUS study, GCNs without pooling layers were able to attain good performance (45).

# E   Additional Ablation Studies

## E.1   Number of Ricci Flow iterations

We investigate the impact of using a varying number of Ricci flow iterations on the performance of ORC-POOL. In the case of node clustering, we re-run ORC-POOL on the Cora and CiteSeer graphs, using 2, 4, and 6 iterations of Ricci flow. In the results presented in the main text, 4 iterations were used.

Table 9: Amazon

| Model | NMI | Time/epoch (s) | Total time (s) |
|---|---|---|---|
| **Diff** | $\mathbf{0.0014 \pm 0.00036}$ | 0.1218 | $53.13 \pm 18.47$ |
| **Mincut** | $0.00004 \pm 0.00013$ | 1.4494 | $364.66 \pm 1.89$ |
| **DMoN** | $0.0012 \pm 0.00009$ | 0.0898 | $35.48 \pm 0.47$ |
| **TV** | $0.0010 \pm 0.00011$ | 0.0980 | $52.99 \pm 16.34$ |
| **ORC (us)** | $0.00067 \pm 0.00077$ | 1.5369 | $405.69 \pm 25.41$ |

Table 10: We report average accuracy (NMI) on Cora and CiteSeer when using varying Ricci flow iterations. The average time per epoch (in seconds) is given in brackets. The reported times (mean/ standard deviation) are computed based on 10 trials.

| Iterations | Cora | CiteSeer |
|---|---|---|
| 2 | $0.45 \pm 0.04$  $(0.033 \pm 0.001)$ | $0.34 \pm 0.03$  $(0.028 \pm 0.000)$ |
| 4 | $0.47 \pm 0.04$  $(0.035 \pm 0.000)$ | $0.35 \pm 0.04$  $(0.029 \pm 0.001)$ |
| 6 | $0.44 \pm 0.05$  $(0.034 \pm 0.001)$ | $0.33 \pm 0.02$  $(0.029 \pm 0.001)$ |

We find that the NMI is comparable for different numbers of Ricci flow iterations. This indicates that the performance of  ORC-Pool is not very sensitive to the number of iterations used, and that the curvature-adjustment captures crucial multi-scale structure with only a few iterations.

We also investigate the impact of different numbers of iterations of Ricci flow on a graph classification task. We compared different numbers of iterations for the REDDIT-BINARY data set, testing 2, 3, and 4 iterations. In the main results, 3 iterations were used. Our results show comparable results across all experiments, implying that the observations in node-level tasks above extend to this setting. Based on our results, we expect that the performance of ORC-Pool is not very sensitive to the number of iterations of Ricci flow and computing the curvature-adjustment based on a small number of iterations already leads to significant performance increases. This is consistent with previous findings for curvature-based methods, see, e.g., (54).

Table 11: We report accuracy (NMI) and time per epoch on REDDIT-BINARY when using varying Ricci flow iterations. The reported times (mean/ standard deviation) are computed based on 10 trials.

| Iterations | Accuracy | Seconds/epoch |
|---|---|---|
| 2 | $0.86 \pm 0.02$ | $11.28 \pm 0.19$ |
| 3 | $0.88 \pm 0.02$ | $12.90 \pm 0.17$ |
| 4 | $0.87 \pm 0.03$ | $13.72 \pm 0.18$ |

### E.2   Choice of Base Layer

For all of the experiments above, we used standard GCN layers as base layers for each experiment. Here, we investigate the performance of our model architecture using GAT and GIN instead of GCN.

For node clustering, we re-run our CiteSeer experiment, again comparing ORC-Pool against four baselines. The experimental setup follows the description in Apx. D.2 apart from GIN and GAT replacing GCN. Our results show a high NMI and overall similar performance gains using ORC-Pool. For graph classification, we make analogous observations when re-running our PROTEINS experiment with GAT and GIN replacing GCN base layers in the model architecture described in Apx. D.3.

### E.3   Comparison with CurvPool (47)

We compare the results of  ORC-Pool and CurvPool on PROTEINS and IMDB-BINARY, using the architecture, hyperparameters, and experimental setup of the CurvPool paper. (47) presents three versions

Table 12: We report accuracy (NMI) on CiteSeer when using GAT and GIN in place of GCN base layers. The average time per epoch (in seconds) is given in brackets. The reported times (mean/ standard deviation) are computed based on 10 trials.

|  | GAT | | GIN | |
| --- | --- | --- | --- | --- |
| **Diff** | $0.28 \pm 0.031$ | $(0.062 \pm 0.004)$ | $0.29 \pm 0.025$ | $(0.097 \pm 0.003)$ |
| **Mincut** | $0.32 \pm 0.018$ | $(0.083 \pm 0.002)$ | $0.33 \pm 0.023$ | $(0.118 \pm 0.004)$ |
| **DMoN** | $0.31 \pm 0.063$ | $(0.077 \pm 0.008)$ | $0.30 \pm 0.070$ | $(0.094 \pm 0.010)$ |
| **TV** | $0.32 \pm 0.016$ | $(0.086 \pm 0.007)$ | $0.30 \pm 0.026$ | $(0.095 \pm 0.008)$ |
| **ORC (us)** | $\mathbf{0.34 \pm 0.040}$ | $(0.156 \pm 0.016)$ | $\mathbf{0.35 \pm 0.040}$ | $(0.219 \pm 0.019)$ |

Table 13: We report average classification accuracy for PROTEINS with GAT and GIN replacing GCN base layers. We use a 80/10/10 train/val/test split. Averages are determined based on 10 trials. Highest accuracy in bold.

|  | GAT | | GIN | |
| --- | --- | --- | --- | --- |
| **Diff** | $0.73 \pm 0.017$ | $(3.82 \pm 0.55)$ | $0.73 \pm 0.022$ | $(4.41 \pm 0.56)$ |
| **Mincut** | $0.76 \pm 0.019$ | $(4.16 \pm 0.67)$ | $\mathbf{0.77 \pm 0.034}$ | $(4.91 \pm 0.64)$ |
| **DMoN** | $0.74 \pm 0.030$ | $(4.83 \pm 0.53)$ | $0.75 \pm 0.070$ | $(5.17 \pm 0.71)$ |
| **TV** | $0.76 \pm 0.028$ | $(4.01 \pm 0.47)$ | $0.76 \pm 0.034$ | $(4.96 \pm 0.048)$ |
| **ORC (us)** | $\mathbf{0.79 \pm 0.022}$ | $(7.10 \pm 0.63)$ | $\mathbf{0.77 \pm 0.039}$ | $(9.21 \pm 0.73)$ |

of CurvPool that differ in their pooling scheme. Below, we compare against the highest performing one, HighCurvPool.

Table 14: We report average accuracy (NMI) for Planetoid for both ORC-Pool and CurvPool.

| **Layer** | Cora | CiteSeer | PubMed |
| --- | --- | --- | --- |
| **Curv** | 0.35 | 0.32 | 0.15 |
| **ORC (us)** | $\mathbf{0.47 \pm 0.04}$ | $\mathbf{0.35 \pm 0.04}$ | $\mathbf{0.24 \pm 0.04}$ |

For node clustering, we compare the two models on three Planetoid data sets. The CurvPool clustering is deterministic, as it is based only on the edge curvatures of the original graph (hence no variance is reported). We used HighCurvPool with a threshold of $-0.2$. We see that ORC-Pool outperforms CurvPool on all three Planetoid graphs, by an especially large margin for Cora and PubMed. This is expected, since ORC-Pool utilizes information encoded in both node attribute and connectivity, whereas CurvPool only utilizes connectivity.

For graph classification, we compared ORC-Pool and CurvPool on one attributed (PROTEINS) and one unattributed data set (IMDB-BINARY). We observe that ORC-Pool achieved higher performance than CurvPool on PROTEINS and equal performance on IMDB-BINARY. This illustrates that ORC-Pool is able to utilize crucial information encoded in the node attributes and that this is crucial for the performance gains observed across data sets.

We note that achieving high performance with CurvPool requires careful tuning of the curvature threshold for pooling nodes. As shown in (47), the choice of this hyperparameter can cause accuracy to change by as much as 5-10 percentage points, depending on the dataset. Therefore, hyperparameter tuning, e.g., grid search needs to be performed to determine the optimal threshold. In contrast, the key hyperparameter in ORC-Poolis the choice of the number of Ricci flow iterations to which downstream performance is fairly insensitive, as the experiments above indicate.

### E.4 Comparison with additional Pooling Layers

We provide an additional comparison with recently introduced pooling layers HoscPool (16), MVPooL (66), GMTPool (6). We follow the previously described experimental setup (see Apx. D.2 and D.3 for details). Our

Table 15: We report average classification accuracy for PROTEINS (attributed) and IMDB-BINARY (unattributed) for ORC-POOL and CURVPOOL. We use 10-fold cross validation. Averages are computed over the test sets for each of the folds. Highest accuracy in bold.

| | PROTEINS | IMDB-BINARY |
|---|---|---|
| **Curv** | $0.77 \pm 0.061$ | $\mathbf{0.71 \pm 0.051}$ |
| **ORC (us)** | $\mathbf{0.79 \pm 0.053}$ | $\mathbf{0.71 \pm 0.049}$ |

experiments show that ORC-Pool achieves the highest accuracy in two graph-level tasks (Mutag, Proteins) and the second highest in a node-level task (Cora). For the latter, only HoscPool achieves a higher accuracy.

| | Cora | MUTAG | Proteins |
|---|---|---|---|
| GMTPool | $0.42 \pm 0.04 \ (0.013 \pm 0.001)$ | $0.86 \pm 0.07 \ (0.066 \pm 0.001)$ | $0.76 \pm 0.04 \ (0.40 \pm 0.00)$ |
| MVPool | $0.32 \pm 0.03 \ (0.015 \pm 0.001 \ )$ | $0.83 \pm 0.08 \ (0.071 \pm 0.001)$ | $0.74 \pm 0.05 \ (0.42 \pm 0.01)$ |
| HoscPool | $\mathbf{0.51 \pm 0.05} \ (0.020 \pm 0.001)$ | $0.87 \pm 0.05 \ (0.089 \pm 0.002)$ | $0.76 \pm 0.04 \ (0.58 \pm 0.01)$ |
| ORC-Pool (us) | $0.47 \pm 0.04 \ \ (0.035 \pm 0.000)$ | $\mathbf{0.90 \pm 0.06} \ (0.190 \pm 0.003)$ | $\mathbf{0.78 \pm 0.04} \ (1.32 \pm 0.00)$ |

Table 16: Comparison of ORC-Pool with additional baselines.

# F   Results on Open Graph Benchmark

We include additional experiments on large-scale data sets from the Open Graph Benchmark (33). For node clustering, we use OGBN-ARXIV, and for graph classification, we use OGBG-MOLHIV.

Like Cora and CiteSeer, OGBN-ARXIV is a citation network. It has 169,343 nodes and 1,166,243 edges; the goal is the classification of papers by subject area. OGBG-MOLHIV is a molecular property prediction dataset. It has 41,127 graphs with an average of 25.5 nodes. We follow the preferred evaluation metrics for OGB, reporting accuracy for OGBN-ARXIV and ROC-AUC for OGBG-MOLHIV. When running ORC-POOL on OGBN-ARXIV, 4 iterations of Ricci flow were used. For OGBG-MOLHIV, 2 iterations of Ricci flow were used.

Table 17: We report accuracy and time per epoch on OGBN-ARXIV for all models.

| | Accuracy | Seconds/epoch |
|---|---|---|
| **No Pool** | $0.59 \pm 0.037$ | $0.33 \pm 0.002$ |
| **Diff** | $0.65 \pm 0.031$ | $0.59 \pm 0.002$ |
| **Mincut** | $0.70 \pm 0.027$ | $3.26 \pm 0.009$ |
| **DMoN** | $0.68 \pm 0.028$ | $0.42 \pm 0.001$ |
| **TV** | $0.68 \pm 0.040$ | $0.41 \pm 0.001$ |
| **ORC (us)** | $\mathbf{0.72 \pm 0.036}$ | $9.29 \pm 0.011$ |

Table 18: We report ROC-AUC and time per epoch on OGBG-MOLHIV for all models.

| | ROC-AUC | Seconds/epoch |
|---|---|---|
| **No pool** | $0.64 \pm 0.038$ | $12.37 \pm 0.78$ |
| **Diff** | $0.68 \pm 0.052$ | $16.87 \pm 1.51$ |
| **Mincut** | $0.70 \pm 0.040$ | $18.38 \pm 2.43$ |
| **DMoN** | $\mathbf{0.73 \pm 0.037}$ | $17.44 \pm 0.89$ |
| **TV** | $0.69 \pm 0.034$ | $17.58 \pm 0.79$ |
| **ORC (us)** | $\mathbf{0.73 \pm 0.048}$ | $34.90 \pm 2.16$ |

We observe that ORC-POOL performs best on OGBN-ARXIV; on OGBN-MOLHIV DMoNPooL performs best. As in the experiments above, these results are based off 10 runs.

# G  Details on Theoretical Analysis

## G.1  Properties of ORC-Pool

**Expressivity.**  In the main text we stated the following result on the impact of ORC-POOL layers on the expressivity of the GNN:

**Corollary 2** (Expressivity of ORC-POOL). *Consider a simple architecture with a block of MP base layers, following by a ORC-POOL layer. Let $G_1, G_2$ denote two 1-WL-distinguishable graphs with node attributes $X_1, X_2$. Further let $X_1' \neq X_2'$ denote the node representations learned by the block of MP layers. Then the coarsened graphs $G_1^P, G_2^P$ learned by the ORC-POOL layer are 1-WL distinguishable.*

*Proof.* This result is a simple adaptation of a recent result by (10, Thm.1), which establishes conditions under which pooling layers preserve expressivity:

**Theorem 4** ( (10), Thm. 1). *Let $G = \{X, E\}$ ($X \in \mathbb{R}^{N \times m}$) denote the input graph and $G'$ the graph obtained after applying a block of MP base layers to $G$; $X'$ denoting the new multiset of node features. Let* POOL: $G' \to G^P$ *denote an SRC pooling layer after the MP layers, which produces a pooled graph $G^P$ with multi-sets $X^P$.* POOL *preserves the expressivity of the MP layers, provided that the following conditions hold:*

1. *Let $G_1, G_2$ denote two WL-distinguishable graphs and $X_1', X_2'$ the node representations learnt by the MP layers. Then $\sum_{x \in X_1'} x \neq \sum_{\tilde{x} \in X_2'} \tilde{x}$.*

2. *The selection function assigns nodes to a unqiue supernode; i.e., $\sum_{j=1}^K s_i^j = 1$ for all $i \in [N]$.*

3. *The reduction function assigns supernode representations as $x_j^P = \sum_{i=1}^N x_i' s_i^j$.*

The authors show that conditions (2),(3) hold for dense pooling layers, such as MINCUTPOOL and, hence, also ORC-POOL. In particular, by construction, graph cuts or partition-based community detection algorithms produce non-overlapping communities; hence the selection function assigns nodes to a unique cluster, which becomes a supernode. The reduction function computes supernode attributes as $X^P = S^T X'$, which aligns with condition (3). Condition (1) is independent of the choice of pooling operator and is fulfilled for any MP layer that is as powerful as the 1-WL test, e.g., GIN (63). □

**Remark 1.** *We note that the inherent limitations in representational power in the MP layers may affect the quality of the coarsening learned by the pooling layer for certain classes of graphs. However, the empirical results presented in this work and the related literature indicate that, in practice, pooling layers coarsen graphs effectively.*

**Permutation-invariance.**  In the main paper, we stated the following property:

**Lemma 2.** *ORC-POOL is permutation-invariant.*

This property is inherited from MINCUTPOOL. For completeness, we provide a the argument below.

*Proof.* Again, we utilize the SRC framework to describe the ORC-POOL operator. The seletion function SEL computes a cluster assignment matrix $S$. It is easy to see that the graph cut objective (Eq. (3.5)) is permutation-equivariant. The permutation-equivariance is not impacted by the curvature-adjustment, as it simply performs a re-weighting of the edges. It is further easy to see that the reduction and connection functions (RED and CON) are permutation-invariant. With that, the composition (SEL ∘ RED) ∘ CON is permutation-invariant, as desired. □

**Pooling and Over-squashing.**  As the number of layers in an MPGNN increases, one observes the formation of bottlenecks (2): Information from far distant nodes is encoded into fixed-length node representations during message passing, leading to information loss. This effect is particularly strong for bridges between clusters, which connect dissimilar nodes whose neighborhoods have a small intersection. Previous literature

has characterized oversquashing via discrete Ricci curvature (40; 20): Edges that connect nodes in different clusters (*bridges*), have low Ricci curvature. This effect is emphasized under ORC flow, as the weight of the bridges increases. With that, our proposed curvature adjustment may amplify oversquashing effects. Recent literature has proposed *graph rewiring* as a mitigation for oversquashing effects. During rewiring, edges are re-sampled proportional to a relevance score (edge weight), which may be assigned utilizing the Lovasz bound (3), random edge dropping (46) or discrete curvature (40; 20), among others. Since we already compute discrete curvature during pooling, we could add a rewiring step with little overhead. We leave an investigation of this approach for future work.

## G.2 Structural Impact of Curvature Adjustment

In the main text, we have given a brief argument for how coarsening approaches based on ORC-POOL preserve the structural integrity of the graph. In this section, we expand on these results. To corroborate our claim that ORC flow reveals coarse geometry and emphasizes the community structure, we employ a graph model, which exhibits community structure as found in real data:

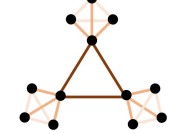

Figure 9: $G_{a,b}$ ($a = b = 3$)

**Definition 2.** *Consider a class of model graphs $G_{a,b}$ ($a > b > 2$), which are constructed by taking a complete graph with $b$ nodes and replacing each by a complete graph with $a + 1$ nodes.*

Graphs of the form $G_{a,b}$ are special cases of stochastic block models (1) with $b$ blocks (*communities*) of size $a + 1$ (see Fig. 9 for an example). Due to the regularity of the graph, we can categorize its edges into three types: (1) bridges, which connect dissimilar nodes in different clusters; (2) internal edges, which connect similar nodes in the same cluster that are at most one hop removed from a dissimilar node; and (3) all other internal edges, which connect similar nodes within the same cluster. (41) derived an evolution equation for edge weights under ORC flow for $G_{a,b}$ graphs:

**Lemma 3** ((41)). *Let $w^t = [w_1^t, w_2^t, w_3^t]$ denote the weights of edges of types (1)-(3). Assuming initialization $w^0 = [1, 1, 1]$, the edge weights evolve under ORC flow $w_{u,v}^{t+1} \leftarrow (1 - \kappa_{uv} d_G(u, v)) w_{u,v}^t$ as*

$$w^{t+1} = \underbrace{\begin{pmatrix} \frac{a-1}{a+b} & \frac{2a}{a+b} & 0 \\ \frac{b}{a+b} & \frac{ab-a-b}{a(a+b)} & \frac{1}{a+b} \\ 0 & 0 & \frac{1}{a} \end{pmatrix}}_{=:F(a,b)} w^t . \tag{9}$$

Building on this characterization, (41) show the follow result, which expresses edge weights $w^t$ with respect to the eigenvalues of the matrix $F_{a,b}$:

**Lemma 4** ((41), adapted to our assumptions). *The matrix $F_{a,b}$ has three real eigenvalues, $\lambda_1, \lambda_2, \lambda_3$, for which $\lambda_1 > 1$, $\lambda_2 = \frac{1}{a}$ and $\lambda_3 < 0$, and corresponding eigenvectors $w_1, w_2, w_3$. Then one can show*

$$w^t = a_1 \lambda_1^t w_1 + a_2 \lambda_2^t w_2 + a_3 \lambda_3^t w_3 = \begin{pmatrix} a_1 \lambda_1^t + o(\lambda_1^t) \\ k a_1 \lambda_1^t + o(\lambda_1^t) \\ \left(\frac{1}{a}\right)^t \end{pmatrix} ,$$

*where $k \in (0, 1)$ and $\lim_{t \to \infty} \frac{o(\lambda_1^t)}{\lambda_1^t} = 0$.*

Utilizing this result we want to analyze the strength of the community structure subject to curvature-adjustment, employing modularity (24) as a measure:

$$Q(W) := \frac{1}{\sum_{uv} w_{uv}} \sum_{uv} \left( w_{uv} - \frac{d_u d_v}{\sum_{uv} w_{uv}} \right) \delta(C_u, C_v) . \tag{10}$$

Here, $W = (w_{ij})$ denotes a weighted adjacency matrix, $d_v$ weighted node degrees and $\delta(C_u, C_v) = 1$, if $u, v$ are in the same cluster and zero otherwise. It can be shown that a higher modularity corresponds to a "stronger" separation of the graph into subgraphs, which is easier to detect; e.g., the corresponding min-cut problem is easier to solve. We show that modularity increases, as edge weights evolve under ORC flow:

**Theorem 5.** *Let $W$ denote the adjacency matrix of the input graph and $C_t$ the curvature-adjusted adjacency matrix after $t$ iterations of ORC flow; i.e., a matrix whose entries correspond to $w_j^t$ for edges of type $j$. Then $Q(C_t) > Q(W)$, with $Q(C_t)$ increasing in $t$ (for $t$ not too small).*

*Proof.* By construction, a $G_{a,b}$ graph has $\frac{b(b-1)}{2}$ type (1), $ab$ type (2) and $\frac{a(a-1)}{2}b$ type (3) edges. Using Lemma 3, we have in iteration $t$ (up to $\frac{o(\lambda_1^t)}{\lambda_1^t}$ error):

$$\sum_{uv} w_{uv}^t \approx \frac{b(b-1)}{2}w_1^t + abw_2^t + \frac{a(a-1)}{2}bw_3^t = \frac{b(b-1)}{2}a_1\lambda_1^t + abka_1\lambda_1^t + \frac{a(a-1)}{2}b\left(\frac{1}{a}\right)^t =: \Sigma^t \;,$$

as well as degrees

$$d_b^t = (b-1)w_1^t + aw_2^t = (b-1)a_1\lambda_1^t + aka_1\lambda_1^t$$

$$d_i^t = w_2^t + aw_3^t = ka_1\lambda_1^t + \left(\frac{1}{a}\right)^{t-1}$$

for internal nodes (subscript $i$) and bridge nodes (subscript $b$), the latter being adjacent to edges of type 1. We can write the modularity of a $G_{a,b}$ graph with weights evolved after $t$ iterations of Ricci flow as

$$Q(C_t) \approx \frac{1}{\Sigma^t}\left(\frac{b(b-1)}{2}\left(w_1^t - \frac{\left(d_b^t\right)^2}{\Sigma^t}\right)\underbrace{\delta(C_u,C_v)}_{=0} + ab\left(w_2^t - \frac{d_i^t d_b^t}{\Sigma^t}\right)\underbrace{\delta(C_u,C_v)}_{=1}\right.$$

$$\left. + \frac{a(a-1)}{2}b\left(w_3^t - \frac{\left(d_i^t\right)^2}{\Sigma^t}\right)\underbrace{\delta(C_u,C_v)}_{=1}\right)$$

$$= \frac{1}{\Sigma^t}\left(ab\left(w_2^t - \frac{d_i^t d_b^t}{\Sigma^t}\right) + \frac{a(a-1)}{2}b\left(w_3^t - \frac{\left(d_i^t\right)^2}{\Sigma^t}\right)\right) \;.$$

Notice that as $t$ increases, we have $d_i^t d_b^t \ll \Sigma^t$ and $\left(d_i^t\right)^2 \ll \Sigma^t$, which implies that the terms $\left(\frac{d_i^t d_b^t}{\Sigma^t}\right)$ and $\left(\frac{\left(d_i^t\right)^2}{\Sigma^t}\right)$ decrease fast with $t$. Moreover, $w_3^t = \left(\frac{1}{a}\right)^t$ is decreasing fast in $t$. Hence, the asymptotics of the sum in brackets are dominated by the term $abw_2^t$. We introduce the shorthands $\Sigma_1^t, \Sigma_2^t, \Sigma_3^t$ for the contributions of edges of types (1)-(3) to $\Sigma^t$. With that the asymptotics of $Q(C_t)$ are dominated by

$$\frac{\Sigma_2^t}{\Sigma_1^t + \Sigma_2^t + \Sigma_3^t} = \frac{1}{1 + \frac{\Sigma_1^t + \Sigma_3^t}{\Sigma_2^t}} \;.$$

We see that

$$\frac{\Sigma_1^t + \Sigma_3^t}{\Sigma_2^t} = \frac{\frac{b(b-1)}{2}\lambda_1^t + \left(\frac{1}{a}\right)^{t-1}}{abk\lambda_1^t}$$

decreases in $t$ with the assumptions in Lemma 3. Putting everything together implies that $Q(C_t)$ increases asymptotically under the Ricci flow. $\square$

### G.3 Licenses

We provide below the licenses of all packages and data sets used in this work.

| Model/Dataset | License | Notes |
|---|---|---|
| LRGB (17) | Custom | See here for license |
| TUDataset (37) | Open | Open sourced here |
| Planetoid (64) | MIT | See here for license |
| Pytorch Geometric (21) | MIT | See here for license |
| Pytorch (44) | 3-clause BSD | See here for license |
| NetworkX (29) | 3-clause BSD | See here for license |
| GraphRicciCurvature (41) | Apache-2.0 license | See here for license |

