# OpenReview forum: "Graph Pooling via Ricci Flow"
_TMLR — Accepted by TMLR_

### Review · Reviewer_YHFn · 2024-11-08

**Summary Of Contributions:**

The paper introduces ORC-Pool, a graph-pooling operator that leverages discrete Ricci curvature to cluster vertices and capture multi-scale graph structures while accounting for both the topology and the vertex attributes of a graph. It further integrates ORC-Pool as a trainable module within Message-Passing Graph Neural Networks. The theoretical properties of the proposed Ricci-Flow-based clustering are formally examined, and experiments demonstrate ORC-Pool’s competitive empirical performance when compared to alternative graph-pooling methods.

**Audience:**

Yes

**Claims And Evidence:**

Yes

**Requested Changes:**

One critical change is a clear discussion on the point raised in the weakness section about what restrictions apply to the GNN-based cluster assignments S.

Beyond this, some additions to the experiments would further strengthen this work:
* It would be helpful to visualize cluster assignments that are actually predicted by a trained GNN model.
* It would also be useful to add results for GNNs without any pooling as a reference baseline in Table 2.

**Strengths And Weaknesses:**

**Strengths**:
* The paper presents novel ideas on how to incorporate topological information into graph pooling operations.
* Extensive theoretical background and analysis of Ricci-Flow-based node clusterings are provided.
* The paper is well-written and easy to follow.


**Weaknesses**:

The most significant weakness is a lack of discussion of potential limitations when predicting cluster assignments end-to-end with message-passing GNNs. When integrating ORC-Pool into a GNN, the cluster assignment $S$ is computed by an MLP that is applied to a vertex embedding $\tilde{X}$ which is computed by a message-passing GNN:

$S = MLP(\tilde{X}, \psi)$

If I understand this correctly, this is a row-wise operation that applies the same MLP independently to the embedding $\tilde{X}(v)$ of each vertex $v$. Since $\tilde{X}$ is computed through message passing, there are inherent restrictions that apply to the embeddings $\tilde{X}$ and assignment $S$. In particular, if the GNN has $\ell$ message passing layers, then $\tilde{X}(v)$ is fully determined by the structure of the depth-$\ell$ subtree rooted in $v$. If two vertices $u,v \in V$ have structurally identical $\ell$-hop neighborhoods, then $\tilde{X}(u)=\tilde{X}(v)$ and therefore the corresponding cluster assignments $S(u)=S(v)$ are also identical by construction. This will always hold true regardless of the model parameters and training objective. The assignment $S$ computed by a GNN in this manner may, therefore, have very different properties from assignments that are optimal with respect to equation 7. In particular:
1. The graph from Figure 4 can not be properly clustered by a message-passing GNN since for each vertex $v$, we can find a vertex $u$ from another cluster with an identical local structure (i.e., the same 1-WL color).
2. In a $k$-regular graph we would always have $S(v)=S(u)$ for all $v,u \in V$ regardless of structure and curvature.

Therefore, it is important to differentiate between the properties of a theoretical clustering that is optimal with respect to Ricci Flow and a clustering that is actually computed by a GNN. Currently, the paper lacks a clear discussion of this difference and its implications.

---

### Review · Reviewer_RZnK · 2024-11-08

**Summary Of Contributions:**

This paper introduces ORC-Pool, a novel graph pooling method for graph neural networks (GNNs) that uses Ricci flow for coarsening graphs. The method demonstrates strong performance on standard benchmarks, outperforming other popular pooling techniques.

**Audience:**

Yes

**Claims And Evidence:**

Yes

**Requested Changes:**

1. While this work will be a valuable addition to the literature, the current presentation detracts from clearly communicating its content. Below are some suggestions/recommendations for improving readability:
    * Proofread the manuscript for typographical errors.
    * Be mindful of readers who are not familiar with the topic by, e.g., replacing "SRC framework" on page 2 with "the select-reduce-connect framework (SRC)".
    * End Section 1 with an overview of the paper's organization (e.g., "This paper is organized as follows: ...").
    * Add algorithm boxes or similar to Section 3 to summarize the key components of ORC-Pool.
    * Perhaps move the discussion on graph cuts from Section 3 to Section 2.
    * Add a table to the appendix that summarizes the select, reduce, and connect operations of DiffPool, MinCutPool, DMoN, TVPool, and ORC-Pool.
2. Could you clarify: Are the trainable parameters in Eq. 8 optimized as part of the GNN architecture's end-to-end training by adding the loss function from Eq. 8 to the downstream task loss, or is Eq. 8 minimized separately with gradient propagation through the solution?
3. Could you add the baseline "None," representing training without pooling, to Table 2 for comparison?
4. Could you add a link to an anonymized repository with your code?

**Strengths And Weaknesses:**

## Strengths

* **Novelty and Relevance**: The proposed method is new and highly relevant to the graph neural network and geometric deep learning communities.
* **Technical Contribution**: The work is technically sound, with some theoretical properties of the proposed method discussed and proved.
* **Performance**: The proposed method outperforms existing pooling methods on standard graph classification tasks.

## Weaknesses

* **Presentation**: The presentation and organization of the paper could be improved to better communicate the contribution of the work. Improving the flow and clarity of the text would make the work more accessible, especially for readers unfamiliar with the topic. (There are also several typographical errors, such as proper name capitalization (e.g., "graclus" on page 6) and inconsistent section and paragraph title capitalization.) The description of ORC-Pool is spread across Sections 2 and 3, making it difficult for readers to follow without reading these sections multiple times.

---

### Review · Reviewer_SpMC · 2024-11-10

**Summary Of Contributions:**

This paper introduces a new graph pooling operator, ORC Pool, which is based on Olliver's Ricci Curvature. The main idea of ORC Pool is to integrate the features associated with the nodes in graphs. This allows for an effective application of ORC Pool in node clustering and graph classification tasks using graph neural networks with ORC Pool similar to min-cut based pooling layers. The experimental results show improved performance with the ORC Pool layer.

**Audience:**

Yes

**Claims And Evidence:**

Yes

**Requested Changes:**

1. The node clustering experiments were on homophilic graphs where the graph pooling has shown good performance. How does ORC Pool compare to other pooling operators on heterophilic graphs? Exploring this could provide valuable insights, and there are heterophilic node classification datasets like Cornell, Texas, Wisconsin, Chameleon, etc.

2. Graclus pooling is closely related to ORC Pool. Including a comparison to graclus pooling would further strengthen the paper.

3. Suggestion on writing improvements to the draft:
* Abstract: "In both settings" - it is not clear what this means as there have been no two settings described before.
* Page 2, Related Work: "The SRC framework" - please provide the expansion before using abbreviation.
* What is "putative clusters"?
* Please add $1_{i\sim j}$ used in equation 4 to notation
* Add definition of $\hat{D}$ in equation 7
* Please add a paragraph on the methods compared in the experiments such as DIFFPool, DMoN, TVPool

**Strengths And Weaknesses:**

### Strengths
Curvature based methods for clustering are explored in non attribute graphs and demonstrate improved performance by capturing the geometrical aspect. It is an interesting idea to integrate this to attributed graphs which are commonly observed. Moreover, the proposed technique is very simple and easy to apply with graph neural networks.

### Weaknesses
* Although the technique is simple, it is computationally expensive for large dense graphs, limiting its practicality.
* The writing is unclear in some places and the presentation can be improved by covering some preliminaries making the draft self contained (see requested changes)

---

### Decision · Action_Editor_xcyb · 2024-12-19

**Recommendation:** Accept as is

**Comment:**

The paper introduces a trainable pooling operator that extending Ricci flow based clustering to attributed graphs. Building on this, the paper presents a pooling layer for incorporating the proposed operator within a message-passing graph neural network. The paper also discusses the structural properties of the pooling operator.

Claims: The key claim is that the provided numerical experiments demonstrate that the introduced pooling layers lead to improvements in several node-based and graph-level inference tasks.

The reviewers agree that the presented experiments are sufficient to support the claim, and this is in line with my assessment. In most of the experiments, a simple GCN is used as the base GNN, although some experiments using GAT and GIN are included in the appendix. All three of these base GNNs are far from the state-of-the-art in terms of performance, so although the proposed pooling layer leads to a significant performance improvement, it is not so clear that this would carry over to scenarios where a more recent GNN is employed.

Audience: All reviewers concurred that the presented work is technically correct and is likely to be of interest to some members of the TMLR audience.

**Audience:**

The findings in the paper will be of interest to some individuals in TMLR's audience.

**Claims And Evidence:**

The claims are supported by accurate, convincing and clear evidence.